# Improved Estimation of O-B Bias and Standard Deviation by an RFI Restoration Method for AMSR-2 C-Band Observations over North America

**Wangbin Shen [1], Zhaohui Lin [2,3], Zhengkun Qin [1,3,*] and Xuesong Bai [1]**

[1] Center of Data Assimilation for Research and Application, Nanjing University of Information Science and Technology, Nanjing 210044, China

[2] International Center for Climate and Environment Sciences, Institute of Atmospheric Physics, Chinese Academy of Sciences, Beijing 100029, China

[3] Collaborative Innovation Center on Forecast and Evaluation of Meteorological Disasters, Nanjing University of Information Science and Technology, Nanjing 210044, China

[*] Correspondence: qzk_0@nuist.edu.cn; Tel.: +86-1879-580-3638

**Abstract:** Spaceborne microwave radiometer observations play vital roles in surface parameter retrievals and data assimilation, but widespread radio-frequency interference (RFI) signals in the C-band channel result in a lack of valuable data over large areas. Establishing repaired data based on existing observation information is crucial. In this study, Advanced Microwave Scanning Radiometer (AMSR)-2 C-band data affected by RFI were accurately repaired through the iterative principal component analysis (PCA) method in 2016 over the U.S. land area. The standard deviation (STD) and bias characteristics of the brightness temperature in the C-band vertical polarization channel were compared and analyzed before and after the restoration to verify the assimilation application prospect of the repaired data. Not only was the spatial continuity of the microwave imager observations significantly improved following restoration; the STD and bias of the observation minus background (OMB) of the restored data were basically consistent with those of the RFI-free data. The STD of OMB exhibited obvious seasonal variations, which were approximately 4.0 K from January to May and 3.0 K from June to December, whereas the biases were near zero in winter but negative (approximately −2.0 K) in summer. The surface type and terrain height also critically affected the STD and bias. The STD decreased with increasing terrain height, whereas the bias exhibited the opposite trend. The STD was largest in low-vegetation areas (4.0 K) but only approximately 2.0–3.0 K in pine forest and brush areas. These results show that the restored data have a high prospect for retrieval application and assimilation, and the STD and bias estimation results also provide a reference for land-based AMSR-2 data assimilation.

**Keywords:** AMSR-2; radio frequency interference; PCA iterative restoration; community radiative transfer model; bias correction

## 1. Introduction

A number of low-frequency microwave radiometers have been put into use (e.g., AMSR-2, Advanced Microwave Scanning Radiometer 2, etc.), which have offered opportunities for the derivation of more direct surface parameter estimations [1–5]. Modern numerical weather predictions (NWPs) rely on assimilating these satellite observations and retrievals to initialize the current state of the land surface accurately [6–11].

The continuous improvement of the assimilation effect has always been the goal of AMSR-2 data assimilation research [12,13]. During the data assimilation process, appropriate adjustment of the background field is determined by the observation error characteristics of the observation data and the background field, as well as some physical mechanisms. Due to the lack of true values, the STD of OMB is often used to characterize the observation

error in data assimilation studies. Therefore, accurate STD estimations of OMB have an essential impact on the effect of data assimilation [14–16].

Bias estimation also plays a crucial role in satellite data assimilation (DA), since it is assumed that the differences between the background and observations satisfy an unbiased Gaussian distribution. In DA theory, systematic bias between satellite-observed and model-simulated radiances should be removed as a necessary condition for meeting this requirement [17,18]. Furthermore, the corrected brightness temperatures are also essential for other steps within DA, for example, cloud detection [19], which depends on the observation-minus-background (OMB) departures [18]. The proper treatment of such systematic biases is critical for the success of data assimilation systems [9,20–26].

Many studies have shown that both effective bias correction and STD estimation are significant prerequisites for successful data assimilation [9,25], but the current estimation methods mostly provide a uniform estimate over the ocean in consideration of the high spatial consistency of the ocean surface. However, the biggest difference between land and sea is the complex underlying surface characteristics of land.

In addition to large STDs caused by the artificial RFI, the variable underlying surface types over land cause considerable error in the surface emissivity. Moreover, a change in surface elevation will further complicate the simulation errors of brightness temperature caused by the surface temperatures and surface emissivity. Therefore, the assimilation of AMSR-2 data over land requires the targeted estimation of OMB standard deviations for different vegetation types and terrain heights on the basis of the current accuracy of the surface emissivity and surface temperature. Thus, the observation weight can be adjusted adaptively in the actual assimilation process and the effective assimilation of the AMSR-2 data over land can be realized.

However, the research on bias correction and STD estimation for AMSR-2 data has been restricted by RFI. AMSR2, which contains a low-frequency C-band (6.9-GHz and 7.3-GHz channels) and an X-band (10.7-GHz channel), is suitable for soil moisture monitoring [27–29]. The optimal low-frequency channel for data assimilation and retrieval using AMSR-2 is the 6.9-GHz channel, as this relatively low frequency responds to a deeper soil layer and is less attenuated by the atmosphere and vegetation than other channels [30]. However, the 6.9-GHz channel is also prone to interference by RFI signals, and the strong signal interference of RFI makes it impossible to effectively estimate the STD and bias of data from this channel, which makes the application of the channel data very difficult. Japan Aerospace Exploration Agency (JAXA) soil moisture products are mainly constructed based on the results retrieved from the 10.7-GHz channel due to the wide range of radio frequency interference (RFI) that occurs globally [28].

RFI refers to the radiation signal received by a satellite microwave radiometer that is confused by active remote sensing signals with similar bands to those used in human activities [31]. The strong signals emitted from these interfering sources conceal relatively weak thermal radiation signals from the Earth–atmosphere system, thus leading to the distortion of observations and causing significant increases in the brightness temperature of the detectors at the low-frequency band [31,32]. Numerous studies have shown that RFI is an extremely vital and nonnegligible factor in low-frequency bands (such as the C-band and the X-band), causing an anomalous bias which affects the application of microwave radiometer data [7,33,34].

An RFI filter has been used before data assimilation in a number of studies [7,9,35]. However, eliminating observational data from the low-frequency channel, which is affected by interference, inevitably causes a large amount of data to be wasted, and may also lead to a large range of observation data being lost.

To compensate for the loss of a large amount of observation data caused by RFI, Shen et al. (2019) [36] proposed an RFI data restoration method based on principal component analysis (PCA), making full use of the channel correlation and the spatial continuity of observations.

Most of the studies on AMSR-2 assimilation directly discard the data affected by RFI. Although the restored data can fill a wide range of observational data gaps, the applicability of these restored data in the assimilation process still requires further evaluation; specifically, answering the question of whether this restoration method can retain the STD and bias characteristics of the observational data is crucial for research on the follow-up of targeted bias-corrections and observational weight settings in the assimilation process. Therefore, in this paper, we used the established PCA iterative restoration method to repair RFI-affected data and then evaluated the bias and STD characteristics before and after the restoration process for different vegetation types and terrain heights. We hoped to provide more accurate bias and STD estimation results for AMSR-2 data assimilations over land.

The paper is structured as follows. In Section 2, we briefly describe the AMSR2 radiance data and the community radiative transfer model (CRTM), and give a brief introduction to the RFI detection and restoration method. In Section 3, we presents the validation of the restoration method and then compare and analyze the bias characteristics of the data before and after RFI restoration. Conclusions and discussions are summarized in Section 4.

## 2. Materials and Methods

### 2.1. AMSR-2 Brightness Temperature Observations

AMSR-2, an instrument carried on GCOM-W1, is a 14-channel, dual-polarization conically scanning passive microwave radiometer with 7 frequencies ranging from 6.9 to 89.0 GHz. This radiometer detects faint microwave emissions from the surface and atmosphere of Earth. The AMSR2 radiance observations frequencies are 6.9, 7.3, 10.65, 18.7, 23.8, 36.5, and 89.0 GHz, as listed in Table 1 [37]. The low-frequency channels below 10.65 GHz are usually used to retrieve various surface parameters, such as the soil moisture, vegetation water content, and snow thickness, as they are window channels with strong vegetation- and soil-penetrating abilities [2,3,5]. The surface incident angle of AMSR2 is maintained at 55 degrees, as this angle is less affected by sea surface winds and produces a large difference between the horizontal and vertical polarization results. The interval between the two conical scans is 1.5 s. The satellite advances approximately 10 km along the running track during this interval, and the width of one scanning line is approximately 1450 km. This scanning process can cover 99% of the world in two days.

**Table 1.** AMSR2 characteristics and performance.

| Channel | Frequency (GHz) | Polarization | Bandwidth (MHz) | Resolution (km) | Sensitivity (K) |
|---------|----------------|--------------|-----------------|-----------------|-----------------|
| 1/2 | 6.925 | H/V | 350 | $35 \times 62$ | 0.34 |
| 3/4 | 7.3 | H/V | 350 | $34 \times 58$ | 0.43 |
| 5/6 | 10.65 | H/V | 100 | $24 \times 42$ | 0.7 |
| 7/8 | 18.7 | H/V | 200 | $14 \times 22$ | 0.7 |
| 9/10 | 23.8 | H/V | 400 | $15 \times 26$ | 0.6 |
| 11/12 | 36.5 | H/V | 1000 | $7 \times 12$ | 0.7 |
| 13/14 | 89.0 | H/V | 3000 | $3 \times 5$ | 1.2 |

The study domain is the central and southeastern United States ($30°$–$40°$N, $260°$–$285°$W) where C-band AMSR-2 radiance data are seriously affected by RFI. This domain also includes a variety of temperate land cover types with complex topography [38]. Performing the experiments in this domain allowed us to test the impact of the PCA iterative restoration method on changeable surface types and terrain.

To certify that this restoration method had good stability and prospects for data assimilation, it was necessary to obtain a sufficiently vast data sample to conduct RFI identification and restoration. Therefore, in this study we selected the AMSR-2 L1R-class observed brightness temperature data covering the study domain for the one-year period of 2016 (1 January to 31 December).

### 2.2. Background—CRTM Simulations

Three fast radiative transfer models have been applied worldwide: the radiative transfer for TOVS (RTTOV) [39], the community radiative transfer model (CRTM), and the advanced radiative transfer model system (ARMS) [40]. In particular, the newly developed ARMS model can be applied to the assimilation of data from the Fengyun satellites and those sensors not included in existing radiative transfer models [40,41]. The CRTM was developed by the U.S. Joint Center for Satellite Data Assimilation (JCSDA) to provide fast and accurate satellite radiance simulations and Jacobian calculations at the top of the atmosphere under all weather and surface conditions [42]. Only the CRTM model was used in this study. It can be shown that the measured radiance in this case is a weighted average of the atmospheric temperature profile.

Figure 1 showed the weighting functions calculated by the atmospheric profiles over ocean (a), and at altitudes of 1000 (b), 2000 (c) and 3000 (d) meters over land, respectively.

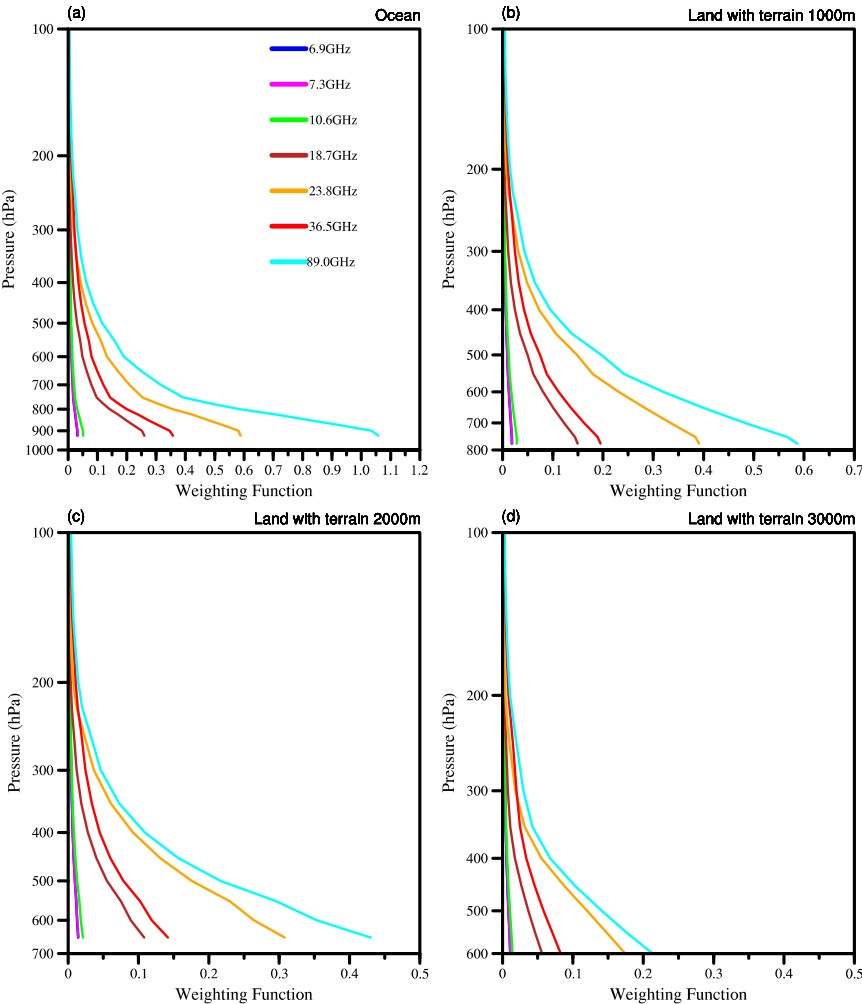

**Figure 1.** Weighting functions of the AMSR-2 channel 1–14 using CRTM based on the atmospheric profile over ocean (**a**) and for terrain height of 1000 m (**b**), 2000 m (**c**) and 3000 m (**d**) over land.

The weighted function $K(p)$ can be calculated as follows:

$$K(p) = d\tau / dln(p) \tag{1}$$

here $\tau$ means the atmospheric transmittance, $p$ is for the pressure [43].

The weighting functions were calculated based on the atmospheric profiles using the CRTM. The profile information includes temperature, specific humidity and pressure

profiles, as well as surface temperature and surface wind field information. It can be seen that weighting functions change little for channels with frequencies less than 10.7 GHz, but for other channels' weighting functions, the differences between the ground and the atmosphere gradually decrease with the increase of terrain height. The weighting functions of the channels with different polarization modes at the same frequency were consistent [44]. The peaks of the weighting function for each channel was located near the surface, as the microwave imager was mainly designed to improve our ability to detect surface parameters through remote sensing.

The amount of radiation detected by the microwave imager is represented by a weighted sum of surface radiation and atmospheric upward microwave radiation in different vertical layers near the ground; this value is mostly sensitive to the atmospheric temperature at the height of the maximum weighting function. The horizontal polarization channel and the vertical polarization channel with the same frequency have the same weighting function.

On the lowest-frequency channel (i.e., 6.9 GHz), the atmosphere contributes the least to the amount of observed radiation. The higher the frequency of the channel is, the wider the weighting function is. The weighting functions of the low-frequency channels are generally located inside the high-frequency channels, except for the 23.8- and 36.5-GHz channels. Thus, the brightness temperatures observed between different channels are highly correlated if the atmospheric contribution is significant [44].

### 2.3. Model Input—ECMWF Reanalysis Data

European Center for Medium-Range Weather Forecasting (ECMWF) hourly reanalysis data, with a horizontal resolution of 0.25 × 0.25 degrees and 37 vertical model levels, were used as the input for the CRTM. The input variables for CRTM include the three-dimensional atmospheric temperature, water vapor mixing ratio, and air pressure, as well as the two-dimensional surface variables of soil moisture, surface skin temperature, wind speed, and wind direction.

Hourly ECMWF liquid water path (LWP) reanalysis data with a horizontal resolution of $0.25° \times 0.25°$ were used to identify data collected under clear-sky conditions.

### 2.4. OMB Calculation Method

In this study, we used the International Geosphere-Biosphere Programme (IGBP) surface type dataset to identify the continental brightness temperature data. Among all the AMSR-2 pixels labeled as "water" in terms of their surface type, further works were carried out to eliminate the pixels within 50 km from coastlines to remove those mixed pixels with water.

Although microwave radiation is able to penetrate some non-precipitating clouds, it is basically unable to penetrate deep precipitation clouds. Even in penetrable clouds, various particles affect microwave radiation through absorption, emission and scattering effects [45,46]. To prevent effects associated with brightness temperature simulation uncertainties in cloudy areas on the bias and STD estimation, in this study we only used data obtained over continental areas under clear-sky conditions.

In order to acquire the simulated brightness temperature at AMSR-2-observed pixel locations and times, polynomial interpolation and linear interpolation were performed on the ECMWF analysis dataset in the horizontal and temporal dimensions, respectively. We processed the hourly ECMWF liquid water path (LWP) data in the same way. The brightness temperature data were considered "cloudy" data when the cloud water path value was greater than 0.01 g/kg, thus allowing us to identify data collected under clear-sky conditions. For the threshold, we referred to the study by Zou et al. (2017) [47]. The total water and ice cloud contents are close to $0.01 \text{ kg m}^{-2}$, which is used as the threshold to detect the cloud in Zou et al. (2017) [47].

Due to the lack of true observed values, the observation errors in the brightness temperature data are mostly estimated by obtaining the standard deviations of the OMB

(observation-minus-background) values [48–51]. In satellite data assimilation, both the observations (*O*) and model simulations (*B*) are assumed to be unbiased. Therefore, STDs can be expressed as:

$$\Delta D_i = O_i - B_i$$

$$\sigma = \sqrt{\frac{\sum_{i=1}^{N}\left(\Delta D_i - \overline{\Delta D}\right)^2}{N-1}} \tag{2}$$

where $O_i$ and $B_i$ are the observed and simulated brightness temperature values on the same pixel, respectively, and $\Delta D_i$ means the OMB value of the pixel. $\overline{\Delta D}$ and $\sigma$ represent the mean value and the standard deviations of the OMB value, respectively. *N* represents the counts of all the continental pixels under clear-sky conditions.

### 2.5. RFI Detection Method—Normalized Principal Component Analysis (NPCA)

The spatial correlations of natural-radiation-generated microwaves among different AMSR-2 instrument observation channels are often very high, as natural surfaces usually produce ultrawideband and smooth microwave radiation.

However, the brightness temperature of the low-frequency AMSR-2 channel increases significantly and abnormally in cases where RFI signals exist, resulting in weakened correlations between these RFI-affected channels and the other channels. The NPCA method, which takes advantage of the aforementioned feature, can effectively identify RFI signals through a PCA decomposition of the constructed interference coefficient matrix, using the brightness temperature difference calculated between the low-frequency channel and the high-frequency channel (low-high). On the other hand, the brightness temperature of the high-frequency channel can be strikingly reduced under the scattering effect of some natural targets (such as ice and snow), thus resulting in an inverse spectral difference gradient in continental regions covered with ice and snow. Therefore, Zou et al. (2013) [52] proposed an RFI detection method for NPCA analyses that has been shown to be effective for identifying RFI in data collected over snow- and ice-covered surfaces; this proposed method is suitable for identifying RFI over complex continental areas with mixed winter snow and RFI signals or over non-scattering surfaces in summer.

### 2.6. RFI Restoration Method—Iterative PCA Method

To compensate for the loss of a large amount of observation data caused by RFI, Shen et al. (2019) [36] proposed an RFI data restoration method based on principal component analysis (PCA). PCA can be used to extract observation information at different spatial scales into some independent PCA modes. The iterative PCA restoration method was established to obtain the correct brightness temperature of the RFI-affected point according to the correct observations around it.

For any observation, if the NPCA method recognizes that this observation has been affected by RFI, then on the satellite orbit where the point is located, the observation data from multiple channels for RFI-free points within the experience range of 350 km around the target point can form a repair matrix containing the target point, but the brightness temperature of the target point will be set to an initial value of 0.

PCA modes representing spatial features with different scales can be obtained through PCA decomposition of the matrix. For any data matrix B, the PCA modes correspond mathematically to the eigenvectors of the covariance matrix of B. The order of the PCA modes is determined based on the eigenvalues of the matrix corresponding to the eigenvectors. The higher-ranked modes correspond to larger eigenvalues, and larger eigenvalues correspond to spatial features with larger values of covariance. In relation to atmospheric variables, a large value of covariance often corresponds to more energy, and the energy of a large-scale weather system is generally much larger than that of a small-scale weather system. Thus, the PCA modes of meteorological variables often correspond to the weather variability features at different scales. More details can be found in Demšar et al. (2013) [53].

The brightness temperature of the target point, determined by means of a large-scale spatial structure, can be obtained by iteratively repeating the reconstruction process of the first mode. The same iterative restoration process can be performed for the rest of the PCA modes, and when all PCA modes are included, the final iterative repair results are obtained.

The proposed restoration method was used to recover observations affected by RFI with high precision [36]. The results of theoretical experiments and real data restoration experiments proved that the accuracy and effectiveness of the new method were much better than those of the Cressman method. Furthermore, the spatial continuity of observations in the recovered data were very well preserved by the new method.

## 3. Results

### 3.1. C-Band Continental RFI Characteristics

The NPCA method, described in Section 2.5, was used for RFI detection on C-band AMSR-2 data in this study. Figure 2 shows the brightness temperatures obtained by the AMSR-2 instrument in the 6.9-GHz and 10.7-GHz vertical polarization channels (Hereinafter referred to as 6.9-GHz-V and 10.7-GHz-V) over the area of the U.S. in the autumn of 2016, as well as the spatial distribution of the RFI signals identified through NPCA (Figure 2a–c). The brightness temperature of the 6.9-GHz channel was generally less than that of the 10.7-GHz channel for most of the continent, because the dielectric constant of water in soil and vegetation depends on this frequency, thus resulting in an increased surface emissivity with an increasing frequency [30]. However, the presence of an RFI signal at the 6.9-GHz frequency caused the brightness temperature of this frequency to increase abnormally, thus resulting in a spectral difference with an opposite sign to that expected. The brightness temperatures of the 6.9-GHz channel in the concentrated areas of Virginia, North Carolina, Texas, and other states were significantly higher than the brightness temperatures of the higher-frequency 10.7-GHz channel, which were far above 300 K, with notable horizontal spatial distribution discontinuities. In the identification results obtained using the NPCA method, the larger the value was, the stronger the possibility of RFI interference. As shown in Figure 2c, regions with abnormally high brightness temperatures (shown in Figure 2a) were detected as having significant RFI signals.

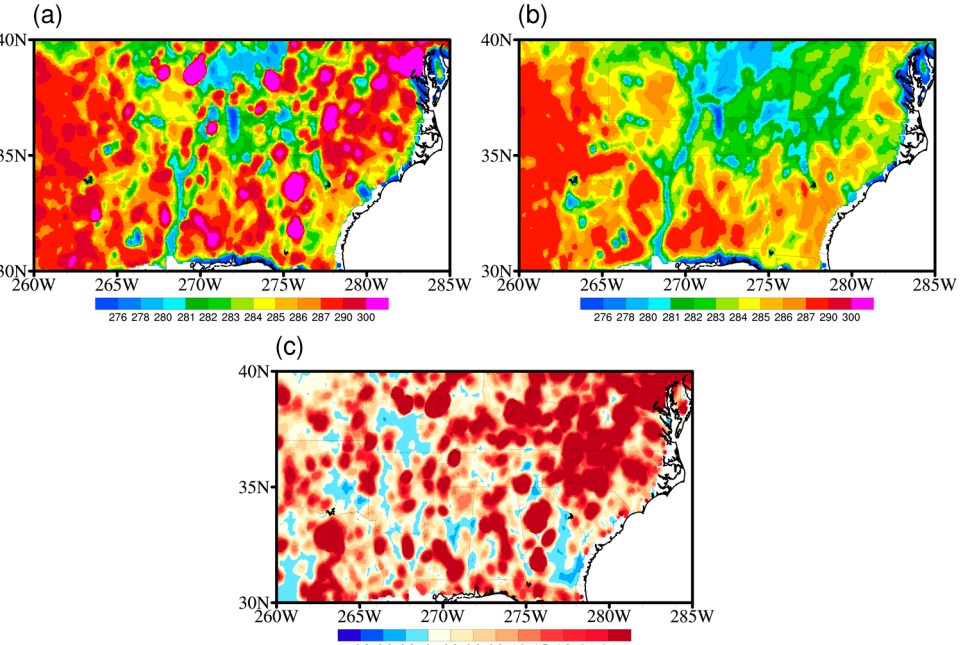

**Figure 2.** Spatial distributions of brightness temperatures of the 6.9-GHz-V channel (**a**) and the 10.7-GHz-V channel (**b**) over the U.S. continental area in the autumn of 2016; RFI signals identified by the NPCA for the 6.9-GHz-V channel are shown in (**c**).

The NPCA method was used for the detection of RFI signals in the horizontal and vertical AMSR-2 6.9-GHz channels over the study domain in 2016, and a daily variation curve of the proportion of the 6.9-GHz-V and 6.9-GHz-H channel scanning points affected by RFI for the land scanning points was obtained for the study domain (Figure 3). In Figure 3, the red line represents the vertical channel and the blue line represents the horizontal polarization channel. The figure shows that both the horizontal and vertical channels in the study region encountered continuous RFI signals throughout the year. In particular, the degree of interference in the vertical channel was obviously greater than that in the horizontal channel. Thirty to forty percent of the data were not available for data assimilation or retrieval applications because of RFI interference.

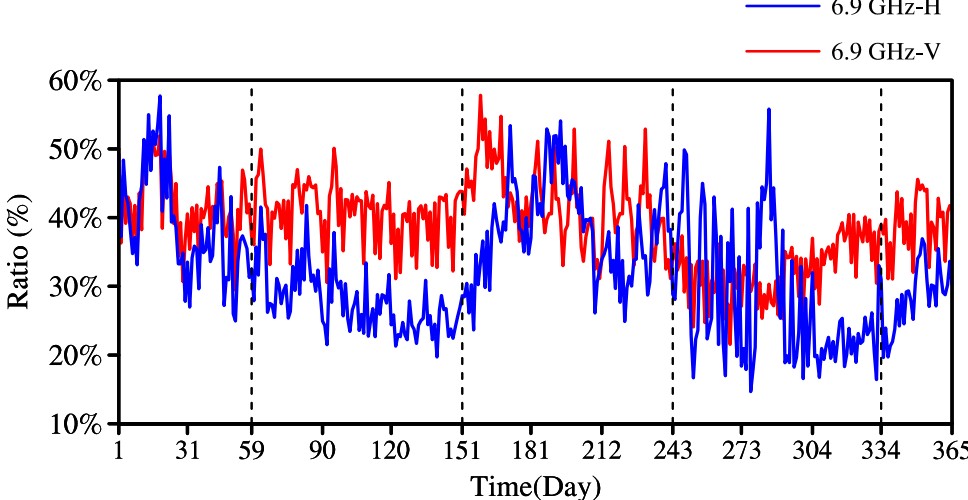

**Figure 3.** Daily variation curves of the proportion of pixels affected by RFI in the study domain for the 6.9-GHz-H (blue) and 6.9-GHz-V (red) channel in 2016.

### 3.2. RFI Restoration and Validation

Figure 4 shows the spatial distributions of the mean observed (a) and restored (b) brightness temperatures of the 6.9-GHz-V channel and the mean observed brightness temperatures of the 7.3-GHz-V (c) and 10-GHz-V (d) channels in autumn 2016. Comparing Figure 4a,b, it can be seen that those abnormally high brightness temperatures caused by RFI were well repaired. The overall geographic distribution of the brightness temperature showed good spatial continuity after this restoration, and the spatial distribution was consistent with the natural surface emission characteristics; in addition, the small-brightness temperature characteristics were restored as well.

In addition to the existing AMSR-E channel, two more channels were added to the AMSR-2 with frequencies near 6.9 GHz and 7.3 GHz. Anne et al. (2015) showed that the RFI phenomenon in the 7.3 GHz observation channel was significantly reduced in the U.S., Japan, and India, where there was severe pollution in the 6.9 GHz channel. As can be seen from Figure 4c, only a few regions showed abnormally high brightness temperatures over 300 K, such as northern West Virginia, central and eastern Alabama, and southern Kansas. However, in the corresponding region of the 6.9-GHz-V channel, there were no abnormally high brightness temperatures. The brightness temperatures of 6.9-GHz-V were generally lower than those of 10.7-GHz-V, except for the RFI-affected region. The frequencies of the 6.9-GHz channel and the 7.3-GHz channel were very close, so the brightness temperatures of the 7.3-GHz channel could be used qualitatively to verify the correctness of the repaired brightness temperatures. It can be seen that the spatial structure of the restored brightness temperature was similar to that of the 7.3-GHz channel. The low-value center in the middle of the region was well reproduced, and the spatial structures of three brightness temperature centers in the northeast of the United States, which were severely impacted by RFI, were also well restored.

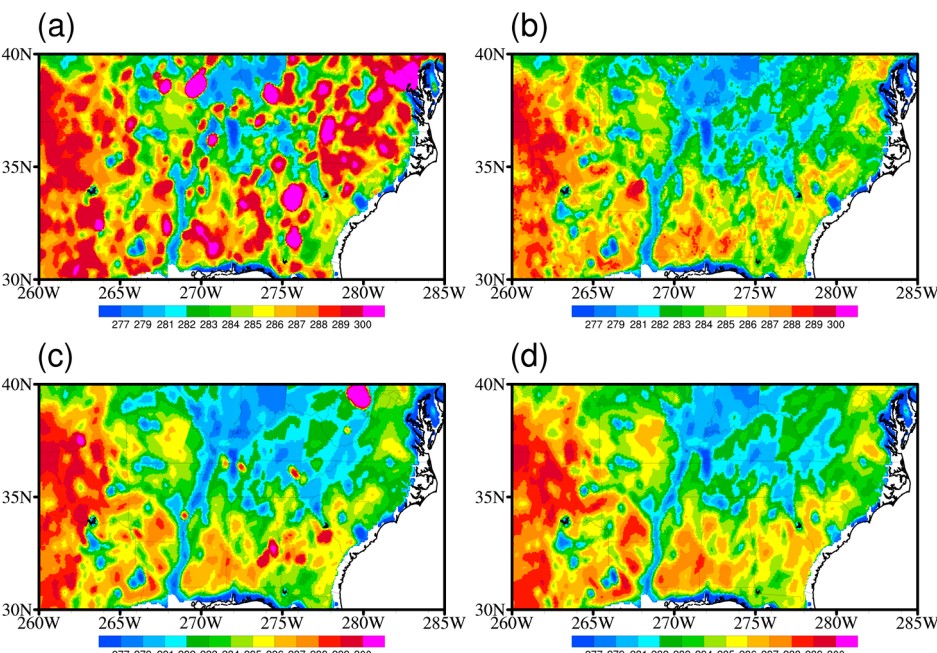

**Figure 4.** Spatial distributions of mean observed (**a**) and restored (**b**) brightness temperatures of the 6.9-GHz-V channel and the observed 7.3-GHz-V (**c**) and 10.7-GHz-V (**d**) channels in autumn 2016.

Figure 5 shows the distribution of the brightness temperature difference between the 6.9-GHz-V channel and the two high-frequency channels, 7.3-GHz-V (a) and 10.7-GHz-V (c), respectively. Figure 5b,d are the same as Figure 5a,c except for the restored brightness temperatures of the 6.9-GHz-V channel. RFI interference led to an abnormal increase in the brightness temperature values, resulting in the opposite spectral differences. Therefore, the larger the positive value in the spectral difference, the more affected were the values in the 6.9-GHz-V channel by RFI. As can be seen in Figure 5a,c, a large area of this region was affected by RFI, and the differences were even greater than 10 K. As can be seen in Figure 5b,d, this difference was basically within 5 K after the repair process. This indicates that the abnormal brightness temperature was well corrected, and also proves the effectiveness of the restoration method.

In consideration of the relatively high percentage of RFI signals in the 6.9-GHz-V channel (the red curve in Figure 3), in this study, we focused on the observation bias and STDs of the 6.9-GHz-V channel in the subsequent analysis.

### 3.3. Comparison of Simulated Brightness Temperature under Clear- and Cloudy-Sky Conditions over Ocean

The hourly cloud liquid water paths based on ERA5 reanalysis data were used to detect clear sky and cloudy data. In order to prove the accuracy of the cloud detection process and the reliability of CRTM simulation, the ocean surface area within the study area was selected for the comparison of OMB characteristics between clear-sky and cloudy areas.

Here, the monthly OMB standard deviations in clear-sky (blue line) and cloudy-sky (red line) areas were calculated separately (Figure 6). The OMB standard deviation in the cloudy area was approximately 6.0 K; this value was much larger than that obtained for the clear-sky area, with an obvious monthly difference. The largest standard deviation, reaching 7.26 K, was observed in June, whereas the smallest value was obtained for December. This may be due to the prevailing convective weather in summer, resulting in more deep clouds. However, the simulation errors for the clear-sky area were primarily reduced, as the standard deviation was maintained at around 0.9 K with a minimal standard deviation of approximately 0.6 K from June to July. The stationary standard deviation in the clear sky areas also proves the effectiveness of the cloud detection method.

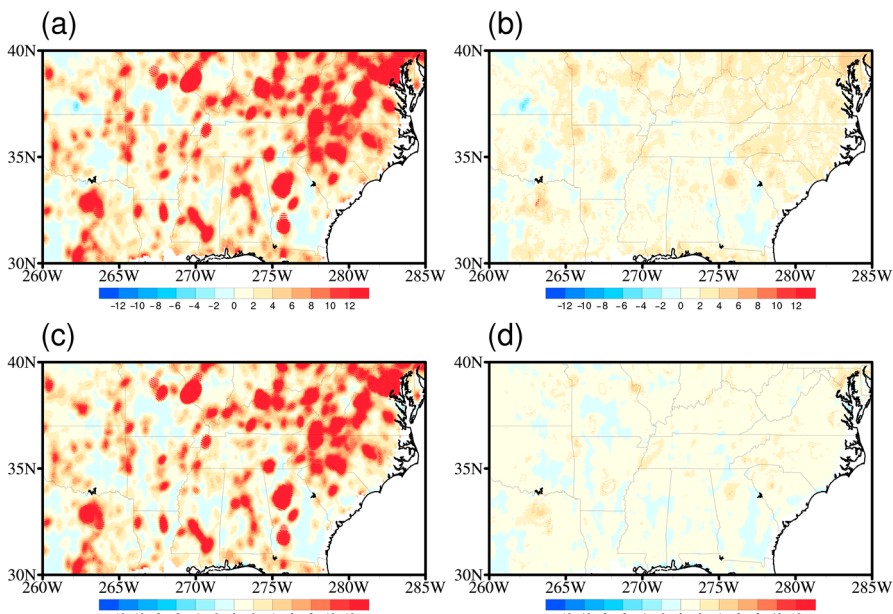

**Figure 5.** Spectral difference between the observed 6.9-GHz-V channel and the 7.3-GHz-V (**a**) and 10.7-GHz-V (**c**) channels, respectively, and the restored 6.9-GHz-V channel and observed 7.3-GHz-V (**b**) and 10.7-GHz-V (**d**) channels in autumn 2016.

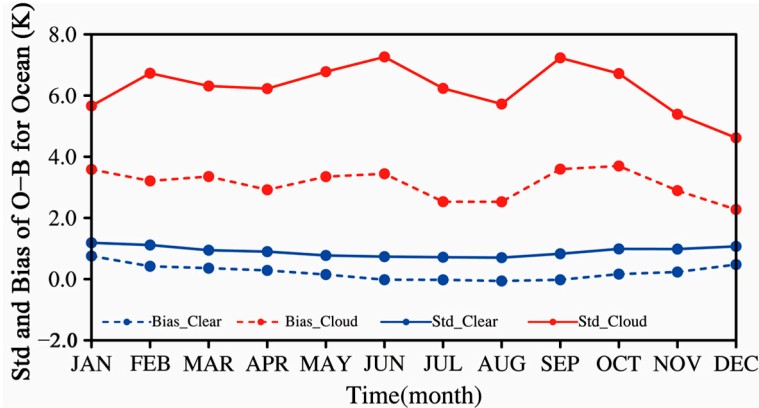

**Figure 6.** Monthly variations of OMB standard deviations (solid line) and bias (dotted line) for data in clear-sky (blue line) and cloudy-sky (red line) conditions over ocean from the 6.9-GHz-V channel in 2016.

It can be seen that there was also a large discrepancy between the monthly OMB bias in clear-sky (blue dotted line) and cloudy-sky (red dotted line) areas over ocean. The simulation was relatively accurate in clear-sky conditions, and the bias was basically below 1 K, with a minimum bias of zero in summer. The bias under cloudy conditions was significantly larger than that for clear-sky areas on the whole, and the bias value was basically around 3 K, with a maximum value of 3.8 K in September and October. The bias changed slightly from January to June, and was stable around 3.7 K.

### 3.4. Standard Deviation over Land

Figure 7 depicts the averaged OMB values before and after the restoration for the 6.9-GHz-Vchannel within the selected domain in autumn 2016. It reveals that the RFI area exhibited an obviously large bias without restoration (Figure 7a), basically exceeding 15.0 K and even exceeding 100.0 K at the maximum point. The simulation errors in the RFI-affected area were significantly reduced following the repair process (Figure 7b), with errors basically within 5.0 K, apart from some systematic deviations in high-terrain areas. Using

the training data sets obtained under RFI-free conditions from AMSR-E, Wu et al. (2011) [54] developed the linear relationship between the measurements obtained at 10.7 GHz and those at 18.7 or 6.9 GHz. Then, the RFI-affected brightness temperatures were corrected based on the RFI-free measurements at 18.7 or 10.7 GHz via this linear relationship. The RFI-correction algorithm was able to produce brightness temperatures at AMSR-E frequencies with a root mean square (RMS) error of no more than 1.5 K. In this study, we focused on the 6.9-GHz-V channel of AMSR-2. The standard deviation of the OMB of this channel was 6.7 K, and it decreased to 4.0 K after restoration using the PCA iterative method.

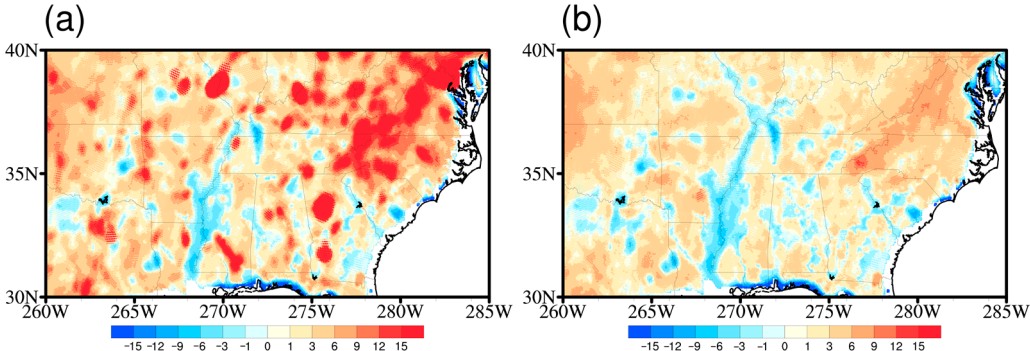

**Figure 7.** Averaged OMB before (**a**) and after (**b**) the restoration for the 6.9-GHz-V channel in autumn 2016.

Although it is clear that the spatial continuity of the brightness temperature data was improved through the restoration process, the impact of this restoration method on the standard deviations still needs to be further clarified in order to apply this method in data assimilation. Figure 8 shows the standard deviation and mean values of the OMB before (magenta line) and after the restoration (red line) for RFI-affected data of the 6.9-GHz-V channel. For comparison, the undisturbed data (blue line) are also shown here. The pink and gray bars in Figure 8a represent the numbers of RFI-affected and RFI-free pixels, respectively. The OMB standard deviation of the unpolluted data was approximately 4.0 K from January to May, whereas this value remained at approximately 3.0 K from June to December. The standard deviation for RFI-interfered data was significantly higher than that of the pollution-free data, with a value of approximately 8.0 K, with a minimal OMB standard deviation of 6.4 K obtained in June. From this OMB standard deviation comparison, it can be seen that the OMB STD values of RFI-affected data were significantly reduced after the restoration. The OMB STD of the restored data in each month was basically similar to that obtained from the RFI-free data; even monthly variation characteristics were also effectively reproduced in these OMB STDs.

As seen from the bias variation shown in Figure 8b, the bias of the RFI-free data was within the range of ±3.0 K. This varied obviously with the season, about 2 K in winter and −2 K in summer. From winter to summer, the bias basically showed the characteristics of a gradual decrease. The bias of RFI-affected data was significantly higher than that of the RFI-free data. The high values reached 9 K, and the low values were above 3 K. It also showed the same seasonal variation characteristics as the correct data. However, after the restoration, the bias derived for each month was very close to that obtained from the nonpolluted data, and the seasonal variation characteristics were effectively reproduced, further confirming the rationality of the restoration method. The land surface temperature had a strong impact on the simulated brightness temperatures. Some previous studies have pointed out that there are obvious seasonal biases in the surface temperature of ERA5 LST, attributed to uncertainty in land surface variables such as the leaf area index and land cover type, etc. [55]. This is a possible reason for the formation of seasonal differences in OMB biases.

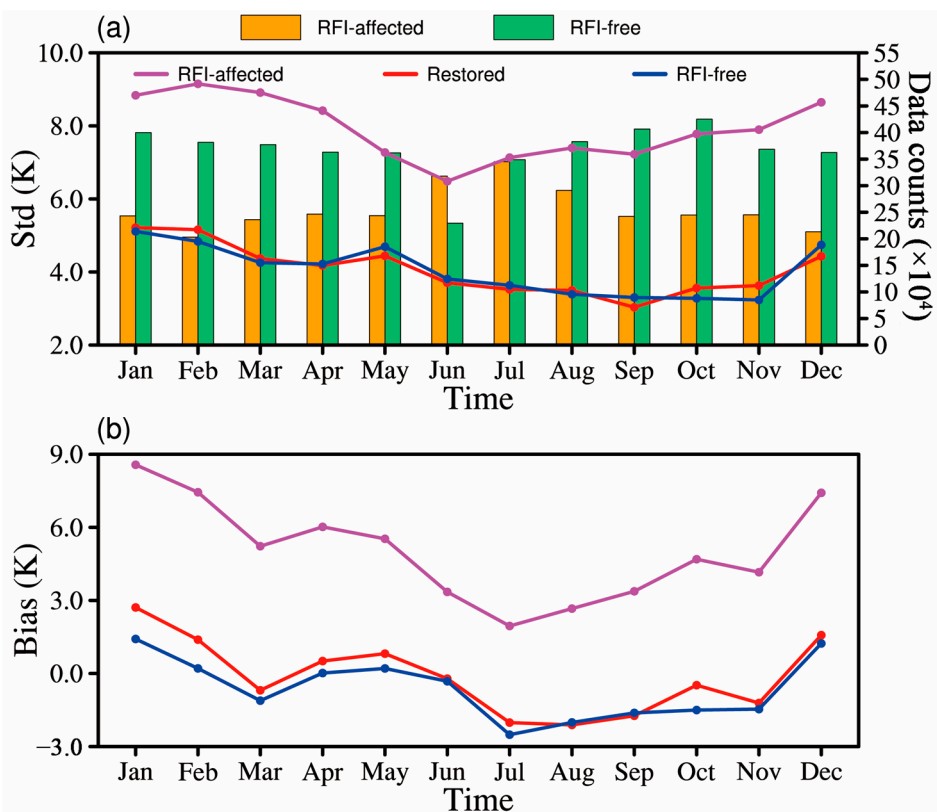

**Figure 8.** Monthly variations of the OMB standard deviation (**a**) and bias (**b**) obtained from the RFI-affected data before (magenta) and after (red) the restoration process and from the RFI-free data (blue) in 2016. The column bars represent the counts of considered data.

### 3.5. Variation Characteristics of STDs with Terrain Height and Surface Type

In contrast with the marine domain, which has uniform underlying surface properties, the underlying surfaces in land areas have two important characteristics: significant discrepancies in topographic height and changeable surface types. The STDs and bias estimation results obtained in land areas are thus inevitably affected by these two characteristics.

The biggest discrepancy between the assimilation of microwave imaging data over the land surface and the ocean is the complexity of the land surface's emissivity. In the microwave range, the land emissivity model is complicated as the land emissivity of each surface type depends on different parameters, such as soil moisture, topography, and the presence and physical properties of vegetation or snow [56]. The surface emissivity error may be significantly different for different land surface types, which will inevitably lead to inconsistency in the brightness temperature simulation bias observed over different land surface types. Therefore, it is necessary to estimate the STDs according to different surface types for the assimilation of AMSR-2 data over land. In addition, the errors of the surface temperature and wind field are much larger than those of variables in the upper atmosphere, so it is particularly important to estimate the OMB bias and STD according to the land cover type. After that, the effective bias correction and observation error specification can be achieved in the assimilation, to effectively account for the observation information of different vegetation types.

To increase the representative of the statistical results, the OMB values of the AMSR-2 6.9-GHz-V channel in the study domain were converted into grid data with a horizontal resolution of $0.25° \times 0.25°$. The spatial distributions of the standard deviations and bias before and after the restoration for 2016 within the analyzed land area are show in Figure 9. For comparison, the spatial distribution of terrain and vegetation types are also shown in the figure.

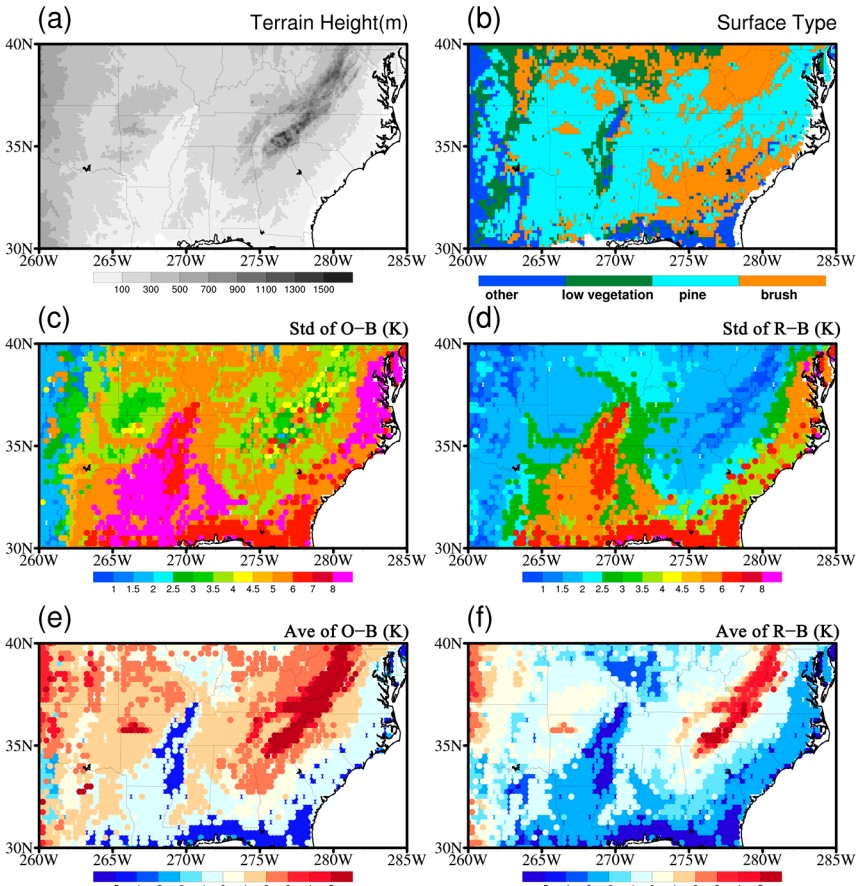

**Figure 9.** Spatial distributions of the terrain heights (**a**), surface types (**b**), standard deviations (**c**,**d**), and bias (**e**,**f**) before (**c**,**e**) and after the restoration (**d**,**f**) in the analyzed land area.

It can be seen from the topographical distribution shown in Figure 9a that the topography in the study domain was complex, exhibiting a large gradient that was mainly characterized by a distribution in which eastern areas were higher than western areas. The elevation of the Appalachian Mountains in the eastern study domain was relatively high, ranging from approximately 1000 to 1500 m. The elevations in the west Mississippi River Plain and the south Gulf Coast Plain were lower in comparison. As seen from the surface-type distribution (Figure 9b), the study domain mainly consisted of distributed pine trees, brush forests, and a small area of low vegetation. As seen from the STD distribution of the integral observed data in the 6.9-GHz-V channel (Figure 9c), which was abnormally large (above 4.0 K), the whole study domain was seriously affected by RFI before the restoration was applied. In the domain, the region with the largest STD—of approximately 7.0 K—was found in the Mississippi River Plain.

After the restoration of the disturbed brightness temperature data, the standard deviations characterizing this region were significantly reduced (Figure 9d). The standard deviation in the Mississippi River Plain area was approximately 3.0 K; this value was basically reduced to approximately 2.0 K in the other areas. The standard deviation in the Appalachian Mountain region basically decreased to less than 1.0 K after the restoration; this value was lower than that of the plain region because the low-frequency AMSR-2 observations are highly sensitive to soil moisture variations, which were relatively small in the mountainous region, leading to the smaller STDs obtained for this area than those obtained for the plains region. The observation bias was correspondingly large due to the strong RFI effect, as seen from its distribution (Figure 9e), with the highest mean value located in the Appalachian Mountains at approximately 8.0 K. The bias in the plain area was relatively low, with values between −3 and 3.0 K. After the restoration, the biases

in most areas decreased significantly, approaching close to 0.0 K, but a positive bias was maintained in high-terrain areas, whereas the negative bias persisted in coastal and central low terrain areas.

Figure 9 shows that the spatial distribution of bias and standard deviations was very similar to that of vegetation types. In order to further clarify the impact of RFI restoration on different vegetation types, Figure 10 presents the OMB mean values and standard deviations before and after the restoration of RFI-affected data under different vegetation types.

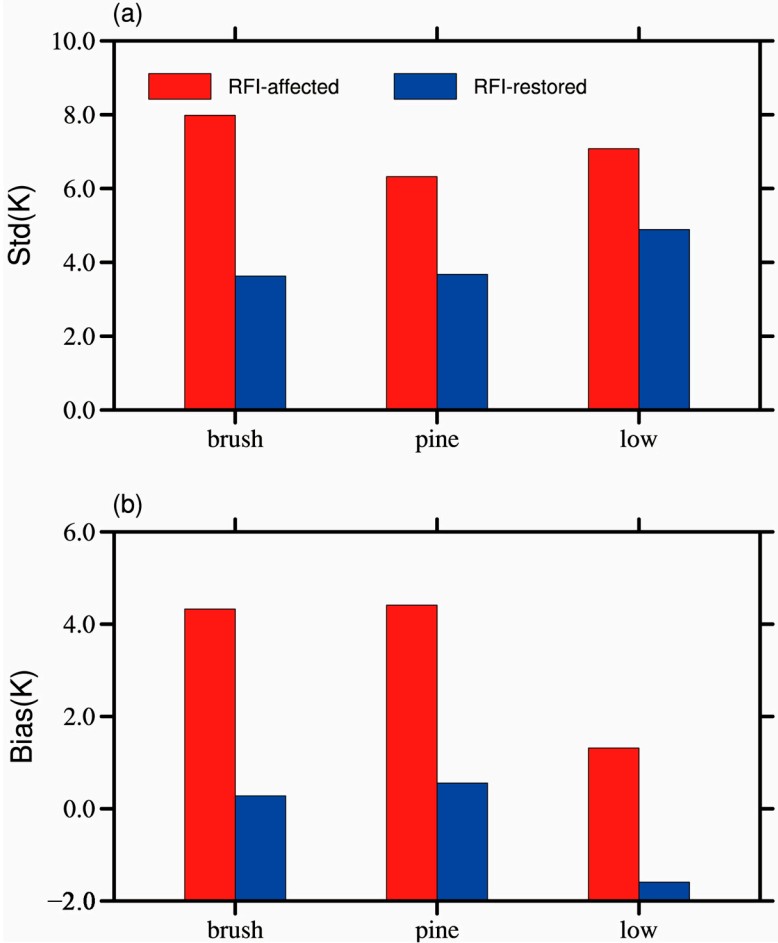

**Figure 10.** Comparison of standard deviation (**a**) and mean values (**b**) of OMB before (red bar) and after (blue bar) the restoration of RFI-affected data for the AMSR-2 6.9-GHz-V channel in 2016 over brush, pine forest, and low vegetation within the selected domain.

It can be seen in Figure 10a that STDs were obviously reduced after RFI restoration under all different surface types. Among these, the restoration effect of brush-covered area was the most significant, with the STD decreasing from 8.0 K to about 3.6 K. Furthermore, the STD decreased from 6.3 K to about 3.6 K within pine-forest-covered areas. The STD of low-vegetation regions was the highest after restoration, around 4.8 K. This is because increased vegetation cover and surface roughness reduce the sensitivity of microwave observations to soil moisture, leading to greater uncertainty in the background simulation [30].

The bias of the restored data was also significantly lower than before. The bias of pine and brush forest regions decreased from around 4.0 K to about 0.0 K. The bias was reduced from 1 K to −2.0 K over low-vegetation area after the accurate repair process.

In addition to vegetation types, the rapidly changing topographic height is another important feature of the land surface that is different from the ocean surface. In order to further

evaluate the characteristics of data errors over land, the bias and STDs are also presented here for data under different terrain heights and different vegetation types (Figure 11). In contrast with Figure 10, the statistics here include all RFI-affected and RFI-free data, so that the statistical results can be directly applied to the actual data assimilation process.

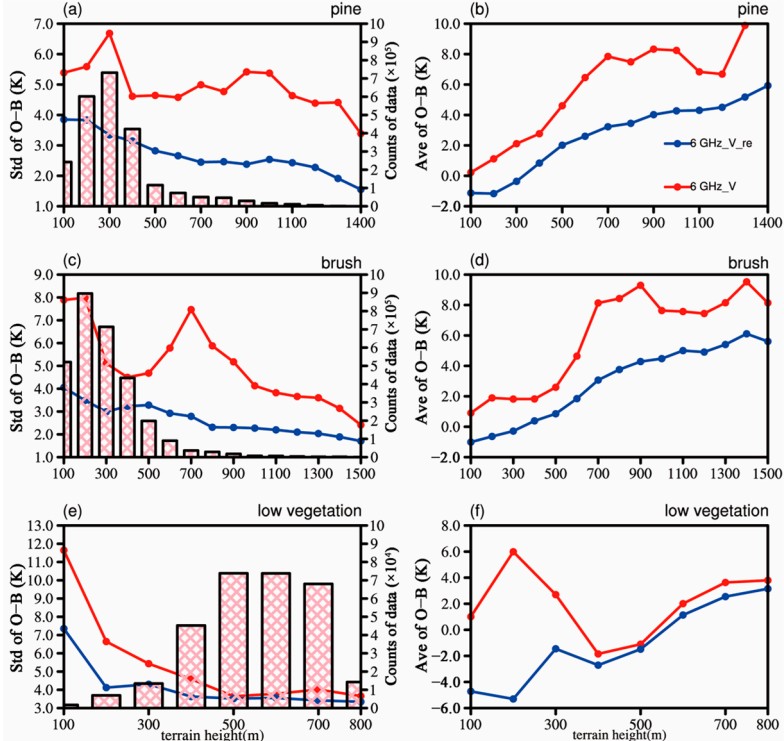

**Figure 11.** Variation curves of the OMB mean values (**b**,**d**,**f**) and standard deviations (**a**,**c**,**e**) characterizing the 6.9-GHz-V AMSR-2 channel with terrain height in the U.S. in 2016. The red and blue lines represent the pre- and post-repair results, respectively.

Figure 11 shows the variation characteristics of the STDs and bias obtained before and after the restoration with varying terrain heights and surface types, with pink reticulated bars indicating the amount of data processed. In this study, we analyzed three major ground types that corresponded to large amounts of data, namely, pine forests, brush regions, and low vegetation. The results revealed obvious differences in the influence of RFI on the brightness temperatures corresponding to different vegetation types under different terrain heights. Among these differences, in pine-forest- and brush-covered areas, the restoration method had an obvious improvement effect on the STD and bias values at different elevations. The STD even reached 8.0 K before the restoration, whereas this value was maintained between 2.0 and 3.0 K following the restoration, decreasing gradually with increasing terrain. The bias value obviously increased with increasing terrain height; this trend was contrary to that of the STDs. The bias value increased rapidly with increasing terrain height below 700 m. When the elevation reached heights above 700 m, the bias was reduced from 8 K to basically below 4.0 K overall following the repair process. In the area covered by low vegetation (Figure 11e,f), RFI was most serious at elevations located below 500 m, where the STD even reached 12.0 K; this value decreased to approximately 4.0 K following the restoration. In the areas with elevations over 500 m, the STDs obtained before and after the restoration were similar, both of which were approximately 4.0 K, and these barely changed with regard to terrain variations. The maximum bias obtained for elevations below 500 m before restoration was 6.0 K, and this value was gradually stabilized from −4.0 K to −2.0 K following the restoration process.

## 4. Discussion

The data obtained from microwave radiometer observations have important application value, especially in the case of low-frequency-channel observations, which play a crucial role in the surface parameter retrieval and data assimilation required in NWP; however, the effects of large-range RFI signals in these low-frequency channels lead to a large amount of observation data being wasted.

To obtain more effective observational data that are applicable to retrieval and assimilation tasks, an iterative PCA method was proposed to repair the RFI-affected data. Although it is clear that the spatial continuity of the brightness temperature data was improved through the restoration, the question of whether this restoration method can retain the STD and bias characteristics of the observational data is crucial for subsequent targeted bias-correction and observational weight-setting research in data assimilation.

Based on AMSR-2 observations from 1 January 2016, to 31 December 2016, in this study, we used the NPCA method to identify RFI-affected data on the C-band (6.9 GHz) in the central and southeastern United States and then applied an iterative PCA method to repair the corrupted data.

Finally, the STD and bias characteristics of the data obtained before and after the repair task and of the pollution-free data collected from the 6.9-GHz-V channel were analyzed in detail, and specifically, the variation characteristics of the STD and bias observed in land areas with varying terrain heights and surface types were further examined, thus providing a corresponding reference for subsequent data assimilation tasks involving low-frequency-channel data from AMSR-2 in land areas.

The long-term restoration results obtained herein show that the applied restoration method was not affected by the terrain height, vegetation type, or seasonal differences. Therefore, the next step will involve assimilating the restored brightness temperatures into numerical models to explore the impacts of the brightness temperature restoration process on the data assimilation.

## 5. Conclusions

In this study, RFI-affected AMSR-2 C-band data regarding the U.S. land area in 2016 were accurately repaired through iterative principal component analysis (PCA). The STD and bias characteristics of the brightness temperature data in the C-band vertical polarization channel were compared and analyzed before and after restoration to verify the assimilation potential of the repaired data. The main conclusions of this work are described below.

(1) The NPCA method was used to identify RFI signals in the observed brightness temperature data representing the U.S., collected from the 6.9-GHz channel for 2016. The results showed that severe RFI impacts persisted throughout the year in the U.S. The interference sources were mainly distributed in areas containing cities, such as the states of Virginia, North Carolina, and Texas. The amount of data suffering from RFI accounted for approximately 40% of the total amount of analyzed data.

(2) Based on the iterative PCA method applied herein, the disturbed brightness temperatures throughout the year were repaired. On the whole, the abnormally high brightness temperatures corresponding to RFI areas were repaired with a high level of precision. The overall brightness temperature distribution conformed to natural surface emission characteristics, maintaining good spatial continuity following the repair process, with small-scale features also being effectively recovered. At the same time, the applied restoration method was not affected by seasonal changes in brightness temperature or by variations in terrain or vegetation types and thus exhibited good stability and prospects for long-term RFI data recovery.

(3) The STD and bias in RFI-affected areas were significantly reduced following the restoration process; in addition, both of them were consistent with the corresponding values obtained from the pollution-free data, indicating that the repaired data retained the bias and STD characteristics of the observation instrument. Furthermore, in pine-forest-



and brush-covered areas, the restoration method had an obvious improvement effect. Over land, the STD decreased gradually with increasing terrain, but the trend of the bias was the opposite. These findings will be useful for subsequent data assimilation applications.

**Author Contributions:** Conceptualization, Z.Q. and W.S.; methodology, W.S.; software, W.S.; validation, Z.Q., W.S. and Z.L.; formal analysis, Z.L.; investigation, X.B.; resources, Z.Q.; data curation, W.S.; writing—original draft preparation, W.S.; writing—review and editing, Z.Q.; visualization, W.S.; supervision, Z.L.; project administration, Z.Q.; funding acquisition, Z.Q. All authors have read and agreed to the published version of the manuscript.

**Funding:** This research was jointly funded by the National Key R&D Program of China (Grant 2018YFC1507302), the National Natural Science Foundation of China (Grant 42075166), the Youth Project of the National Natural Science Foundation of China (41805076), the Natural Science Foundation of Jiangsu Province (BK20211396) and the FengYun-3 meteorological satellite engineering ground application Project (FY-3(03)-AS-11.08).

**Data Availability Statement:** ERA5 hourly reanalysis data: https://cds.climate.copernicus.eu/cdsapp#!/search?type=dataset&text=era5 (accessed on 2 March 2022); AMSR-2 brightness temperature data: https://gportal.jaxa.jp/gpr/search?tab=0 (accessed on 13 July 2021).

**Acknowledgments:** We would like to acknowledge the suggestions given by reviewers and editor. And we are grateful to the High-Performance Computing Centre of the Nanjing University of Information Science and Technology for the performed numerical calculations in this study using its blade cluster system.

**Conflicts of Interest:** The authors declare no conflict of interest.

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
