# Peer review of "Improved Estimation of O-B Bias and Standard Deviation by an RFI Restoration Method for AMSR-2 C-Band Observations over North America"

_remotesensing, doi:10.3390/rs14215558_

Round 1
Reviewer 1 Report
Review of the manuscript
«Improved Estimation of Observation Errors by an RFI Restoration Method for AMSR-2 C-band Observations over North American» by
Wangbin Shen, Zhaohui Lin, Zhengkun Qin, and Xuesong Bai
Assessing the accuracy of brightness temperature retrieval from space data remains an urgent task. The manuscript presents interesting scientific results. It corresponds to the topic of MDPI Remote Sensing in the section Active and passive microwave remote sensing.
Recommendations for the authors:
1. In Introduction, the statement of the problem should be formulated more clearly and differentiated from the problems solved by other researchers. In addition, the novelty of the study should be discussed in terms of the choice of territory and research method. For a potential reader, it is important to show what tasks have already been solved and what gap in the knowledge is solved by the reviewed manuscript.
2. The manuscript should compare the accuracy of restored data with the results reported in other similar studies. Quantitative characteristics should be given.
3. In my opinion, the authors incorrectly use the term "error". According to modern ideas in metrology about this term, it is necessary to provide a confidence interval when calculating the measurement error. The authors do not give a confidence interval, but actually investigate the experimental standard deviation. Thus, the manuscript deals with the measurement uncertainty, rather than the measurement error. The authors should revise properly the title and text of the manuscript.
4. The term “experimental standard deviation” or “standard deviation” should be used in place of the term “observation error”.
5. In my opinion, the correct form of Equation (1) for the experimental standard deviation is .
6. In figure captions, the authors should provide explanations for Figs. 4a, 4b, 4c; 7a, 7b (Page 8, Lines 284, 285; Page 20, Lines 345-347).
7. The line “Figure 9. Comparison of Standard Deviation (A) and Mean Values (b) ...” lacks gaps after the words “deviation” and “values” (Page 12, Line 405).
After the proper revision, the manuscript can be published in MDPI Remote Sensing.

Author Response
Review of the manuscript
«Improved Estimation of Observation Errors by an RFI Restoration Method for AMSR-2 C-band Observations over North American» by
Wangbin Shen, Zhaohui Lin, Zhengkun Qin, and Xuesong Bai
Assessing the accuracy of brightness temperature retrieval from space data remains an urgent task. The manuscript presents interesting scientific results. It corresponds to the topic of MDPI Remote Sensing in the section Active and passive microwave remote sensing.
Recommendations for the authors:
- In Introduction, the statement of the problem should be formulated more clearly and differentiated from the problems solved by other researchers. In addition, the novelty of the study should be discussed in terms of the choice of territory and research method. For a potential reader, it is important to show what tasks have already been solved and what gap in the knowledge is solved by the reviewed manuscript.
Reply:
Thanks for reviewer's suggestions. The following discussions have been added to the introduction:
Most of the studies on AMSR-2 assimilation directly discard the data affected by RFI currently. There are few researches on the restoration of RFI interfered data. Although the proposed restored data method can fill a wide range of observational data gaps, the applicability of these restored data in the assimilation process still requires further evaluation.
On the other hand, the current estimation methods mostly provide a uniform estimate over the ocean in consideration of the high spatial consistency of the ocean surface. But the variable underlying surface types over land cause considerable error in the surface emissivity. Moreover, an increase in terrain height will further complicate the simulation errors of brightness temperature caused by the surface temperatures and surface emissivity. Therefore, the assimilation of AMSR-2 data over land requires the targeted estimation of OMB standard deviations for different vegetation types and terrain heights on the basis of the current accuracy of the surface emissivity and surface temperature. Thus, the observation weight can be adjusted adaptively in the actual assimilation process and the effective assimilation of the AMSR-2 data over land can be realized.
And we added the relevant discussion as follows in line 41 of the introduction,:
“During the data assimilation process, appropriate adjustment of the background field is determined by the observation error characteristics of the observation data and the background field, as well as some physical mechanisms. Due to the lack of true values, the STD of OMB is often used to characterize the observation error in data assimilation studies. Therefore, accurate STD estimations of OMB have an essential impact on the effect of data assimilation (Geer and Bauer, 2010; Yang et al., 2016; Tandeo et al., 2020).”
And these following contents are added in line 107:
“Many studies have shown that both effective bias correction and STD estimation are significant prerequisites for successful data assimilation (Lievens et al., 2016; Han and Niels, 2016), but the current estimation methods mostly provide a uniform estimate over the ocean in consideration of the high spatial consistency of the ocean surface. However, the biggest difference between land and sea is the complex underlying surface characteristics of land.”
“In addition to large STDs caused by the artificial RFI, the variable underlying surface types over land cause considerable error in the surface emissivity. Moreover, an increase in terrain height will further complicate the simulation errors of brightness temperature caused by the surface temperatures and surface emissivity. Therefore, the assimilation of AMSR-2 data over land requires the targeted estimation of OMB standard deviations for different vegetation types and terrain heights on the basis of the current accuracy of the surface emissivity and surface temperature. Thus, the observation weight can be adjusted adaptively in the actual assimilation process and the effective assimilation of the AMSR-2 data over land can be realized.”
- The manuscript should compare the accuracy of restored data with the results reported in other similar studies. Quantitative characteristics should be given.
Reply:
Thanks for reviewer's suggestions. We added a comparison with the results of Wu et al. (2011), and the relevant discussion was added in line 1459 of the revised article:
“Using the training data sets under RFI-free conditions of AMSR-E, Wu et al. (2011) developed the linear relationship between the measurements at 10.65 GHz and those at 18.7 or 6.925 GHz. Then the contaminated brightness temperatures were corrected from the RFI-free measurements at 18.7 or 10.65 GHz by the linear relationship. The RFI-correction algorithm can produce a brightness temperature at AMSR-E frequencies with a root mean square (RMS) error of no more than 1.5 K. This study focused on the 6-GHz-V channel of AMSR-2. The standard deviation of OMB of this channel was 6.7 K, and it decreased to 4.0 K after restoration by PCA iterative method.”
The article is as follows:
Wu, Y.; Weng, F. Detection and correction of AMSR-E radio-frequency interference. Acta Meteorologica Sinica 2011, 25, 669-681.
- In my opinion, the authors incorrectly use the term "error". According to modern ideas in metrology about this term, it is necessary to provide a confidence interval when calculating the measurement error. The authors do not give a confidence interval, but actually investigate the experimental standard deviation. Thus, the manuscript deals with the measurement uncertainty, rather than the measurement error. The authors should revise properly the title and text of the manuscript.
Reply:
Sorry for the confusion. We have changed the term “error” to “standard deviation”.
The title was revised as: “Improved Estimation of O-B bias and standard deviation by an RFI Restoration Method for AMSR-2 C-band Observations over North America”.
- The term “experimental standard deviation” or “standard deviation” should be used in place of the term “observation error”.
Reply:
We have revised it in manuscript.
- In my opinion, the correct form of Equation (1) for the experimental standard deviation is .
Reply:
The formula from reviewer is correct, and the standard deviation of the paper is also calculated according to this formula, but the formula was incorrectly input in the paper, which has been corrected.
- In figure captions, the authors should provide explanations for Figs. 4a, 4b, 4c; 7a, 7b (Page 8, Lines 284, 285; Page 20, Lines 345-347).
Reply:
Sorry for the missing of explanations. The figure captions of Figure 4 have been changed to “Figure 4. Spatial distributions of mean observed (a) and restored (b) brightness temperatures of the 6-GHz-V channel and the observed 7-GHz-V (c) and 10-GHz-V (d) channels in autumn 2016.”
The figure captions of Figure 7 have been changed to “Figure 8. Monthly variations of the OMB standard deviation (a) and bias (b) obtained from the RFI-affected data before (magenta) and after (red) the restoration process and from the RFI-free data (blue) in 2016. The column bars represent the counts of considered data.”
- The line “Figure 9. Comparison of Standard Deviation (A) and Mean Values (b) ...” lacks gaps after the words “deviation” and “values” (Page 12, Line 405).
Reply:
Sorry for the incorrect format. It has been revised.
After the proper revision, the manuscript can be published in MDPI Remote Sensing.

Reviewer 2 Report
See attached file

Author Response
Thanks for all your suggestions! And you can also find the details in attached files!
“Improved Estimation of Observation Errors by an RFI Restoration Method for AMSR-2 C-band Observations over North American” by Shen et al.
Authors proposed an iterative principal component analysis (PCA) to correct AMSR-2 C-band RFI contaminated data and evaluate the effectiveness of the techniques through both spatial distribution of O-B and the maps before and after the correction. Overall, the technique is interesting and has some potential for uses in satellite data assimilation. However, there are still several outstanding issues that require further clarifications and analyses. Please see the detail comments as follows.
Major comments:
- The O-B of AMSR-2 C-band brightness temperatures over land is impacted by many factors other than RFI recovering process. It is a long wave toward direct radiance assimilation over land. Authors mentioned many times on assimilation of AMSR2 data over land. Please elaborate more how to assimilate the data over land in atmospheric data assimilation system.
Reply:
Thanks for reviewer's suggestions. We added the possible research direction of AMSR-2 data over land in the application of atmospheric data assimilation in line 41 of the introduction, and the relevant discussion is as follows:
“During the data assimilation process, appropriate adjustment of the background field is determined by the observation error characteristics of the observation data and the background field, as well as some physical mechanisms. Due to the lack of true values, the STD of OMB is often used to characterize the observation error in data assimilation studies. Therefore, accurate STD estimations of OMB have an essential impact on the effect of data assimilation (Geer and Bauer, 2010; Yang et al., 2016; Tandeo et al., 2020).”
According to the reviewer's suggestion, we also discussed other factors which impact the O-B of AMSR-2 C-band brightness temperatures over land other than RFI recovering process. Over land areas, the surface emissivity simulation is impacted by the variable underlying surface types and the increase of terrain height, which will further complicate the simulation errors of brightness temperature. And these following contents are added in line 107:
“Many studies showed that both effective bias correction and STD estimation are significant prerequisites for successful data assimilation (Lievens et al., 2016; Han and Niels, 2016), but the current estimation methods mostly give a uniform estimate over ocean in considering of the high spatial consistency of the ocean surface. However, the biggest difference between land and sea is the complex underlying surface characteristics of land.
In addition to large STDs caused by the artificial RFI, the variable underlying surface types over land cause considerable error of the surface emissivity. And the increase of terrain height will further complicate the simulation errors of brightness temperature caused by the surface temperatures and surface emissivity. Therefore, the assimilation of AMSR-2 data over land requires targeted estimation of OMB standard deviations for different vegetation types and terrain heights on the basis of the current accuracy of the surface emissivity and surface temperature. Thus, the observation weight can be adjusted adaptively in the actual assimilation and realize the effective assimilation of the AMSR-2 data over land.”
- The biases are near 0 in winter but negative (approximately -2.0 K) in summer can be explained. What causes this seasonal dependent biases.
Reply:
Thanks for reviewer's suggestions. Trigo et al. (2015)’s study showed that there is obvious seasonal characteristics in the error of land surface temperature (LST) in ERA5 reanalysis data. Therefore, it is reasonable to believe that the seasonal simulation errors of ERA5 LST lead to the seasonal bias difference in this study. The following discussion has been added to line 1613 of the revised article:
“The land surface temperature had a strong impact on the simulated brightness temperatures. Some previous studies have pointed out that there are obvious seasonal biases in the surface temperature of ERA5 LST, attributed to uncertainty in land surface variables such as the leaf area index and land cover type, etc. (Trigo et al., 2015). This is a possible reason for the formation of seasonal differences in OMB biases.”
Relevant literature has been added to the references of the revised article:
Trigo, I.F.; Boussetta, S.; Viterbo, P.; Balsamo, G.; Beljaars, A.; Sandu, I. Comparison of model land skin temperature with remotely sensed estimates and assessment of surface-atmosphere coupling. Journal of Geophysical Research: Atmospheres 2015, 120, 12,096-012,111.
- Also, the standard deviation of O-B is strongly dependent on surface type. Please explain the physical reasons of this phenomenon.
Reply:
Prigent et al., (2011) pointed out that the land emissivity model is complicate as land emissivity of each surface type depends on different land parameters in the microwave range. The surface emissivity error may be significantly different under different land surface types, which will inevitably lead to the inconsistency of brightness temperature simulation bias over different land surface types. The relevant physical explanation is as follows which has also been added to the revised manuscript in line 1630:
“The biggest discrepancy between the assimilation of microwave imaging data over the land surface and the ocean is the complexity of the land surface’s emissivity. In the microwave range, the land emissivity model is complicated as the land emissivity of each surface type depends on different parameters, such as soil moisture, topography, and the presence and physical properties of vegetation or snow (Prigent et al., 2011). The surface emissivity error may be significantly different for different land surface types, which will inevitably leads to inconsistency in the brightness temperature simulation bias observed over different land surface types. Therefore, it is necessary to estimate the STDs according to different surface types for the assimilation of AMSR-2 data over land. In addition, the errors of the surface temperature and wind field are much larger than that of variables in the upper atmosphere, so it is particularly important to estimate the OMB bias and STD according to the land cover type. After that, the effective bias correction and observation error specification can be achieved in the assimilation, so as to effectively absorb the observation information of different vegetation types.”
Relevant literature has been added to the references of the revised article:
Prigent, C.; Liang, P.; Tian, Y.; Aires, F.; Moncet, J.L.; Boukabara, S.A. Evaluation of modeled microwave land surface emissivities with satellite‐based estimates. Journal of Geophysical Research: Atmospheres 2015, 120, 2706 - 2718.
- Since there is a new generation of fast radiative transfer model (ARMS) (Weng et al., 2020, AAS) is available. Please explain what is the major advantages of using CRTM over ARMS in terms of AMSR2 simulation.
Reply:
Thanks for reviewer's suggestions. We believe that the ARMS model is good enough for our study, but the ARMS model is not yet available for public download right now, so it was not chosen. The CRTM was developed by the U.S. Joint Center for Satellite Data Assimilation (JCSDA) to provide fast and accurate satellite radiance simulations and Jacobian calculations at the top of the atmosphere under all weather and surface conditions (Weng 2007), then the CRTM model is selected in this study. We added the following related discussion in line 528 of the revised manuscript:
“Three fast radiative transfer models have been applied worldwide: the radiative transfer for TOVS (RTTOV) (Saunders et al., 2018), the community radiative transfer model (CRTM) (Weng et al., 2007), and the advanced radiative transfer model system (ARMS) (Weng et al., 2020). In particular, the newly developed ARMS model can be applied to the assimilation of data from the Fengyun satellites and those sensors not included in existing radiative transfer models. (Weng et al., 2020; Yang et al., 2020). The CRTM was developed by the U.S. Joint Center for Satellite Data Assimilation (JCSDA) to provide fast and accurate satellite radiance simulations and Jacobian calculations at the top of the atmosphere under all weather and surface conditions (Weng 2007). Only the CRTM model was used in this study.”
Relevant literature has been added to the references of the revised article:
Weng, F.; Yu, X.; Duan, Y.; Yang, J.; Wang, J. Advanced Radiative Transfer Modeling System (ARMS): A New-Generation Satellite Observation Operator Developed for Numerical Weather Prediction and Remote Sensing Applications. Advances in Atmospheric Sciences 2020, 37, 131-136, doi:10.1007/s00376-019-9170-2.
Weng, F. Advances in Radiative Transfer Modeling in Support of Satellite Data Assimilation. Journal of The Atmospheric Sciences - J ATMOS SCI 2009, 64, doi:10.1175/2007JAS2112.1.
Saunders, R.W.; Hocking, J.; Turner, E.C.; Rayer, P.J.; Rundle, D.; Brunel, P.; Vidot, J.; Roquet, P.; Matricardi, M.; Geer, A.J., et al. An update on the RTTOV fast radiative transfer model (currently at version 12).
Yang, J.; Ding, S.; Dong, P.; Bi, L.; Yi, B. Advanced radiative transfer modeling system developed for satellite data assimilation and remote sensing applications. Journal of Quantitative Spectroscopy and Radiative Transfer 2020, 251, 107043, doi:https://doi.org/10.1016/j.jqsrt.2020.107043.
- The success of AMSR2 RFI correction should be based on more rigid validation. Brightness temperature spectrum should be checked and compared before and after RFI correction. In addition, the brightness temperature at two C-band channels (6.9, 7.3 GHz) can be also used to evaluate the robustness of the AMSR-2 RFI algorithm.
Reply:
Thanks for reviewer's suggestions. We added the evaluation results of the spectral difference method and the spatial distribution of observed brightness temperature of 7.3 GHz-V channel to further verify the correctness of the restoration data. The relevant results are as follows:
“Fig. 4 shows the spatial distributions of the mean observed (a) and restored (b) brightness temperatures of the 6-GHz-V channel and the mean observed brightness temperatures of the 7-GHz-V(c) and 10-GHz-V(d) channels in autumn 2016. Comparing Fig. 4(a) and Fig.4(b), it can be seen that those abnormally high brightness temperatures caused by RFI were well repaired. The overall horizontal distribution of the brightness temperature showed good spatial continuity after this restoration, and the spatial distribution was consistent with the natural surface emission characteristics; in addition, the small-brightness temperature characteristics were restored as well.
In addition to the existing AMSR-E channel, two more channels were added to the AMSR-2 with frequencies near 6.925 GHz and 7.3 GHz. Anne et al. (2015) showed that the RFI phenomenon in the 7.3 GHz observation channel was significantly reduced in the U.S., Japan, and India, where there was severe pollution in the 6.9 GHz channel. As can be seen from Figure 4c, only a few regions showed abnormally high brightness temperatures over 300 K, such as northern West Virginia, central and eastern Alabama, and southern Kansas. However, in the corresponding region of the 6-GHz-V channel, there were no abnormally high brightness temperatures. The brightness temperatures of 6-GHz-V were generally lower than those of 10-GHz-V, except for the RFI-affected region. The frequencies of the 6-GHz channel and the 7.3-GHz channel were very close, so the brightness temperatures of the 7.3-GHz channel could be used qualitatively to verify the correctness of the repaired brightness temperatures. It can be seen that the spatial structure of the restored brightness temperature was similar to that of the 7.3-GHz channel. The low-value center in the middle of the region was well reproduced, and the spatial structures of three brightness temperature centers in the northeast of the United States, which were severely impacted by RFI, were also well restored.”
Figure 4. Spatial distributions of mean observed (a) and restored (b) brightness temperatures of the 6-GHz-V channel and the observed 7-GHz-V (c) and 10-GHz-V (d) channels in autumn 2016.
“Figure 5 shows the distribution of the brightness temperature difference between the 6-GHz-V channel and the two high-frequency channels, 7-GHz-V (a) and 10-GHz-V (c), respectively. Fig. 5(b) and 5(c) are the same as Fig. 5(a) and Fig. 5(c) except for the restored brightness temperatures of the 6-GHz-V channel. RFI interference led to an abnormal increase in the brightness temperature values, resulting in the opposite spectral differences. Therefore, the larger the positive value in the spectral difference, the more affected were the values in the 6-GHz-V channel by RFI. As can be seen in Fig. 5(a) and 5(c), a large area of this region was affected by RFI, and the differences were even greater than 10 K. As can be seen in Fig. 5(b) and 5(d), this difference was basically within 5 K after the repair process. This indicates that the abnormal brightness temperature was well corrected, and also proves the effectiveness of the restoration method.”
Figure 5. Spectral difference between the observed 6-GHz-V channel and the 7-GHz-V (a) and 10-GHz-V (c) channels, respectively, and the restored 6-GHz-V channel and observed 7-GHz-V (b) and 10-GHz-V (d) channels in autumn 2016.
In addition, we also removed the original Figure 4c, which is the simulated brightness temperature. The main purpose of showing the simulated brightness temperature is to reflect that the spatial continuity of the brightness temperature without the RFI. There is indeed a great discrepancy between the observed and simulated brightness temperature. In order to avoid interfering with the purpose of the study, we deleted Figure 4c.
The following discussions are also removed:
“When comparing with the distribution of the observed brightness temperatures of 7-GHz-V (Fig. 4(c)), it can be seen that both results show similar spatial distributions, with southern areas having higher values than northern areas. However, the simulated brightness temperatures near Mitchell Mountain are lower than the observed values.”
- RFI-uncontaminated should be changed to RFI-free
Reply:
Thanks for reviewer's suggestions. “RFI-uncontaminated” in line 24 have been changed to “RFI-free”.
- Page 3, “The CRTM, which is representative and is currently widely used as an observation operator to simulate the radiative transfer of the satellite-based visible-light, infrared, ultraviolet or microwave channel for satellite-derived data assimilation tasks. “ should be rewritten. It reads awkwardly in both content and the function of CRTM. CRTM can not simulate visible and ultraviolet instruments for satellite data assimilation.
Reply:
Thanks for reviewer's suggestions. We have changed the sentence to:
“The CRTM was developed by the U.S. Joint Center for Satellite Data Assimilation (JCSDA) to provide fast and accurate satellite radiance simulations and Jacobian calculations at the top of the atmosphere under all weather and surface conditions (Weng 2007). Only the CRTM model was used in this study.”
Relevant literature has been added to the references of the revised article:
Weng, F. Advances in Radiative Transfer Modeling in Support of Satellite Data Assimilation. Journal of The Atmospheric Sciences - J ATMOS SCI 2009, 64, doi:10.1175/2007JAS2112.1.
- Figure 1. It is not recommended that simple weighting functions be applied for microwave imagers. Indeed, its weighting function can be highly dependent on the surface emissivity. Please explain more the difference of the weighting function peaks over land and oceans.
Reply:
Thanks for reviewer's suggestions. In order to enrich the information of the weighting function, we showed the weighting functions calculated by the atmospheric profiles over ocean, and at altitudes of 1000, 2000 and 3000 meters over land, respectively. The results are shown in the following figure.
“Fig.1 showed the weighting functions calculated by the atmospheric profiles over ocean (a), and at altitudes of 1000 (b), 2000 (c) and 3000(d) meters over land, respectively. It can be seen that weighting functions changes little for channels with frequencies less than 10.6 GHz, but for other channels’ weighting functions, the differences between the ground and the atmosphere gradually decrease with the increase of terrain height.”
Figure 1. Weighting functions of the AMSR-2 channel 1-14 using CRTM based on the atmospheric profile over ocean (a) and for terrain height of 1000 meters(b), 2000 meters(c) and 3000 meters (d) over land.
- In Figure 4, I can see the huge difference between simulated and observed (after RFI correction) in Rocky mountain regions. An elongated cold brightness temperature zone in observation is not captured in simulation. Please give an interpretation.
Reply:
The possible reason is that the coarse horizontal resolution of the background field causing the difference between water and land surface temperatures is not well reproduced, resulting in a high simulation of brightness temperature in the Mississippi river region. The main purpose of showing the simulated brightness temperature is to reflect the good spatial continuity of brightness temperature without RFI. However, there is indeed a great discrepancy between the observed and simulated brightness temperature. In order to avoid interfering with the purpose of the study, we would like to delete the old Figure 4c.
- Figure 5 only displays the standard deviation of O-B. What is the key point to deliver with this figure. I would also like to see O-B itself.
Reply:
Thanks for reviewer's suggestions. We added the bias feature of O-B, and the revised figure and its relevant discussion has been added is as follows:
“It can be seen that there was also a large discrepancy between the monthly OMB bias in clear-sky (blue dotted line) and cloudy-sky (red dotted line) areas over ocean. The simulation was relatively accurate in clear-sky conditions, and the bias was basically below 1 K, with a minimum bias of zero in summer. The bias under cloudy conditions was significantly larger than that for clear-sky areas on the whole, and the bias value was basically around 3 K, with a maximum value of 3.8 K in September and October. The bias changed slightly from January to June, and was stable around 3.7 K.”
Figure 6. Monthly variations of OMB standard deviations (solid line) and bias (dotted line) for data in clear-sky (blue line) and cloudy-sky (red line) conditions over ocean from the 6-GHz-V channel in 2016.
- Page 11 line 330. The sentence “it can be seen that the utilized repair method has effectively eliminated the influence of RFI” should be rewritten.
Reply:
Thanks for reviewer's suggestions. We have revised it as “it can be seen that the OMB STD values of RFI-affected data were significantly reduced after the restoration.”.
- Page 11, 337, “the bias of the unpolluted data” is bad English.
Reply:
Thanks for reviewer's suggestions. We have changed it into “the bias of the RFI-free data”.
- Figure 8. The label “Obs minus Back” should be changed to O-B for a consistency throughout the paper.
Reply:
Thanks for reviewer's suggestions. We have revised it, the modified figure is as follows:
Figure 8. Spatial distributions of the terrain heights (a), surface types (b), observation error (c-d) and bias (e-f) before (c, e) and after the restoration (d, f) in the analyzed land area.

Reviewer 3 Report
Compared with the article published by the author in 2019, this article only adds the results of the OMB comparison and analysis of the repaired data and the RFI-uncontaminated data, which can be inferred from the verification experiments of the previous article, and lacks scientific significance and innovation. Since this paper claims that the purpose of studying the OMB of the repaired data is to demonstrate the application value of RFI data restoration method in data assimilation, it is suggested that the authors should conduct research to evaluate the impact of the repaired data assimilated into NWP model, and give verification experiments results, which is suitable for publication.
In additional,there are several questions needed to be modified or explained with more details:
1. line 131,weight function for which variable? Atmospheric temperature or humidity?
2. line133,Why are different polarization modes at the same frequency consistent?Please provide references to support this assertion or provide verification results.
3.line147, Why are the brightness temperatures observed between different channels highly correlated if the atmospheric contribution is significant?Please provide references to support this assertion or provide verification results.
4. line177,for the threshold value 0.01kg/kg of cloudy, please supplement reference or provide the basis.
5.line614,references 43-46 are incorrectly formatted
6.line218,why do PCA modes represent spatial features with different scale? Please supplement reference or provide the basis.
7 . line220 and 223 The two sentences are basically repeated.
8. line 284, in Figure 4c, there is a significant difference between the simulated brightness temperature and the repaired observation. There is a low temperature band near 268W in the repaired observation, but not in the simulation. The simulated brightness temperature between 270W-280W is obviously higher than the repaired observation, but between 260W-265W and 35N-40N, the repaired observation is significantly higher than the simulation. What is the assertion that the comparison of simulation and the repaired observation in this paragraph intends to illustrate? Please explain the difference between simulation and the repaired observation.
9 line294,it cannot be seen that the cited reference 48 has relation with this sentence.
Author Response
Thanks for your suggestions! And you can also find the details in attached file!
Compared with the article published by the author in 2019, this article only adds the results of the OMB comparison and analysis of the repaired data and the RFI-uncontaminated data, which can be inferred from the verification experiments of the previous article, and lacks scientific significance and innovation. Since this paper claims that the purpose of studying the OMB of the repaired data is to demonstrate the application value of RFI data restoration method in data assimilation, it is suggested that the authors should conduct research to evaluate the impact of the repaired data assimilated into NWP model, and give verification experiments results, which is suitable for publication.
Reply:
We are sorry that we failed to clearly state the purpose of this study. In our 2019 paper, a new restoration method for RFI interference data is introduced, but only verified the correctness of the restoration method from the probability distribution characteristics of brightness temperature before and after the restoration.
In satellite data assimilation research, we need not only observation data without RFI interference, but also accurate bias and error information of observation data, so as to determine how much observation information is introduced into the background field according to the error and bias characteristics. Therefore, it is necessary to determine whether the restoration data can retain the error and bias characteristics of RFI-free observations, and also tell other researchers whether the restoration data can be applied to data assimilation research.
In addition, most of the current observation bias and error estimates for data assimilation give a single value of bias and error, which is more suitable for satellite data assimilation in the ocean area with uniform surface. Considering that the bias and error of the satellite data in the land area are also affected remarkably by the complex underlying surface type and terrain height, the study also further analyzes the bias and error characteristics under different surface types and terrain heights, and the relevant estimation results can provide more accurate and targeted bias and error information for the assimilation of AMSR-2 data in the land area.
This research mainly focuses on these two purposes. In the subsequent research, we will conduct AMSR-2 data assimilation research in the land area on the basis of this research.
In additional, there are several questions needed to be modified or explained with more details:
- line 131, weight function for which variable? Atmospheric temperature or humidity?
Reply:
Sorry for the confusion. The weighting function is calculated from the atmospheric profile, which including temperature, specific humidity and pressure profiles, as well as surface temperature and surface wind information.
For clarity, we added the following content to revised manuscript in line 539:
“The weighting functions were calculated based on the atmospheric profiles using the CRTM. The profile information includes temperature, specific humidity and pressure profiles, as well as surface temperature and surface wind field information.”
- line133, Why are different polarization modes at the same frequency consistent? Please provide references to support this assertion or provide verification results.
Reply:
Channels with same frequencies but different polarizations have the same weighting function, which is cited from the work of Zou et al. (2012). We also added citations to the article.
The article is as follows:
Zou, X. Introduction to microwave imager radiance observations from polar-orbiting meteorological satellites. Advances in Meteorological Science and Technology 2012, 45-50.
3.line147, Why are the brightness temperatures observed between different channels highly correlated if the atmospheric contribution is significant? Please provide references to support this assertion or provide verification results.
Reply:
The Earth’s surfaces often produce smooth and ultrawideband microwave radiation. So the multichannel correlations of radiometer data from natural radiations are often high. This is cited from the work of Zou et al. (2012). We also added citations to the article.
The article is as follows:
Zou, X. Introduction to microwave imager radiance observations from polar-orbiting meteorological satellites. Advances in Meteorological Science and Technology 2012, 45-50.
- line177,for the threshold value 0.01kg/kg of cloudy, please supplement reference or provide the basis.
Reply:
Thanks for reviewer's suggestions. For the threshold, we referred to the study by Zou et al. (2017). The total water and ice cloud contents are close to 0.01 kg m‒2 which is used as the threshold to detect the cloud in that study.
According to the reviewer's suggestion, we added the following content to the revised manuscript in line 718:
“For the threshold, we referred to the study by Zou et al. (2017). The total water and ice cloud contents are close to 0.01 kg m‒2, which is used as the threshold to detect the cloud in Zou et al. (2017).”
Relevant literature has been added to the references of the revised article:
Zou, X.; Qin, Z.; Weng, F. Impacts from assimilation of one data stream of AMSU-A and MHS radiances on quantitative precipitation forecasts. Quarterly Journal of the Royal Meteorological Society 2017, 143, 731-743.
5.line614,references 43-46 are incorrectly formatted
Reply:
Thanks for reviewer's suggestions. We have revised it.
6.line218,why do PCA modes represent spatial features with different scale? Please supplement reference or provide the basis.
Reply:
Thanks for the reviewer's suggestion. The first few PCA modes correspond to larger eigenvalues, and larger eigenvalues correspond to spatial features with larger value of covariance. For atmospheric variables, large value of covariance often corresponds to more energy, and the energy of large-scale weather variability is generally much larger than that of the small-scale weather variability. So, the PCA modes of meteorological variables often correspond to the weather variability features at different scales. The relevant discussion can be seen in Demšar et al. (2013).
And we added the following content before in line 940:
“For any data matrix B, the PCA modes correspond mathematically to the eigenvectors of the covariance matrix of B. The order of the PCA modes is determined based on the eigenvalues of the matrix corresponding to the eigenvectors. The higher-ranked modes correspond to larger eigenvalues, and larger eigenvalues correspond to spatial features with larger values of covariance. In relation to atmospheric variables, a large value of covariance often corresponds to more energy, and the energy of a large-scale weather system is generally much larger than that of a small-scale weather system. Thus, the PCA modes of meteorological variables often correspond to the weather variability features at different scales. More details can be found in Demšar et al. (2013).”
This document of Demšar et al. (2013) has been cited in the text and has been added to the references.
Urška Demšar , Paul Harris , Chris Brunsdon , A. Stewart Fotheringham & Sean McLoone (2013): Principal Component Analysis on Spatial Data: An Overview, Annals of the Association of American Geographers, 103:1, 106-128.
- line220 and 223 The two sentences are basically repeated.
Reply:
Sorry for the repeat of expressions here. We have changed “The brightness temperature of the target point determined by the large-scale spatial structure can be obtained by using the first mode reconstruction, and the original zero initial value is replaced by the reconstructed brightness temperature. By iteratively repeating the reconstruction process of the first mode, we can obtain the brightness temperature of the target point determined by the large-scale spatial structure.” in line 950 to “The brightness temperature of the target point, determined by means of a large-scale spatial structure, can be obtained by iteratively repeating the reconstruction process of the first mode.”
- line 284, in Figure 4c, there is a significant difference between the simulated brightness temperature and the repaired observation. There is a low temperature band near 268W in the repaired observation, but not in the simulation. The simulated brightness temperature between 270W-280W is obviously higher than the repaired observation, but between 260W-265W and 35N-40N, the repaired observation is significantly higher than the simulation. What is the assertion that the comparison of simulation and the repaired observation in this paragraph intends to illustrate? Please explain the difference between simulation and the repaired observation.
Reply:
The main purpose of the simulated brightness temperature is to reflect that the good spatial continuity of brightness temperature without RFI. There is indeed a great discrepancy between the observed and simulated brightness temperature. In order to avoid interfering with the purpose of the study, we would like to delete the old Figure 4c.
- line294,it cannot be seen that the cited reference 48 has relation with this sentence.
Reply:
Thanks for the reviewer's suggestion. It has been removed.

Round 2
Reviewer 2 Report
I am satisfied with author's responses.
Author Response
Thanks for your suggestions and help for the manuscript!
Reviewer 3 Report
On the one hand, I accept the authors' reply that the major contribution of this paper is to give the OMB bias and error characteristics of RFI restoration data, which is different from the 2019 paper. The results in this paper have some scientific significance and innovation.
On the other hand, for two reasons, I still think that this paper is not a complete study by giving only the OMB bias and error characteristics of the RFI restoration data. First, the AMSR-2 data have been operationally assimilated, so the best way to verify the effectiveness of the RFI restoration method for dada assimilation is still to directly evaluate the assimilation results of the RFI restoration data; second, the reliability of the OMB bias and error characteristics of the RFI restoration data given in the paper still needs to be validated by data assimilation.
Therefore, I suggest that the authors should supplement at least one data assimilation experiment of RFI restoration data using the OMB bias and error characteristics from this paper.
Also, the 1st response is still incorrect. The weight function is obtained by deriving the brightness temperature for some atmospheric variable. Which atmospheric variable corresponds to the weighting function in Figure 1?
Author Response
Review of the manuscript
On the one hand, I accept the authors' reply that the major contribution of this paper is to give the OMB bias and error characteristics of RFI restoration data, which is different from the 2019 paper. The results in this paper have some scientific significance and innovation.
On the other hand, for two reasons, I still think that this paper is not a complete study by giving only the OMB bias and error characteristics of the RFI restoration data. First, the AMSR-2 data have been operationally assimilated, so the best way to verify the effectiveness of the RFI restoration method for dada assimilation is still to directly evaluate the assimilation results of the RFI restoration data; second, the reliability of the OMB bias and error characteristics of the RFI restoration data given in the paper still needs to be validated by data assimilation.
Therefore, I suggest that the authors should supplement at least one data assimilation experiment of RFI restoration data using the OMB bias and error characteristics from this paper.
Reply:
Thanks for reviewer’s suggestions. This research mainly focuses on analyzing that whether the restoration data can retain the error and bias characteristics of RFI-free observations, and also tell other researchers whether the restoration data can be applied to data assimilation research. In the follow-up study, we will evaluate the assimilation effect of the restoration data through the actual assimilation experiments, but these are beyond the scope of this study
Also, the 1st response is still incorrect. The weight function is obtained by deriving the brightness temperature for some atmospheric variable. Which atmospheric variable corresponds to the weighting function in Figure 1?
- Please address the following comment in Report 2 of Reviewer 3 and explain in the manuscript as necessary: The weighting function is obtained by deriving the brightness temperature for some atmospheric variable. Which atmospheric variable corresponds to the weighting function in Figure 1?
Reply:
The weighting functions in Figure 1 represent the vertical gradient of atmospheric transmittance. We added the following description in Line 540:
The weighted function can be calculated as follows:
here τ means the atmospheric transmittance, p is for the pressure (Jean 2003).
Relevant literature has been added to the references of the revised article:
Jean-Noël, T. Satellite data assimilation in numerical weather prediction: an overview. In Proceedings of Seminar on Recent developments in data assimilation for atmosphere and ocean, 8-12 September 2003, Shinfield Park, Reading, 2003.
